# Methods Used for Enhancing the Bioavailability of Oral Curcumin in Randomized Controlled Trials: A Meta-Research Study

**DOI:** 10.3390/ph15080939

**Published:** 2022-07-28

**Authors:** Bruna Mimica, Viljemka Bučević Popović, Ines Banjari, Antonia Jeličić Kadić, Livia Puljak

**Affiliations:** 1Department of Chemistry, Faculty of Science, University of Split, 21000 Split, Croatia; brunamimica1@gmail.com (B.M.); viljemka@pmfst.hr (V.B.P.); 2Department of Food and Nutrition Research, Faculty of Food Technology, University of Osijek, 31000 Osijek, Croatia; ibanjari@ptfos.hr; 3Department of Pediatrics, University Hospital Split, 21000 Split, Croatia; jelicic.antonia@gmail.com; 4Center for Evidence-Based Medicine and Health Care, Catholic University of Croatia, 10000 Zagreb, Croatia

**Keywords:** curcumin, bioavailability, clinical trial

## Abstract

It is unknown how randomized controlled trials (RCTs) approach the problem related to curcumin bioavailability. We analyzed methods and reporting regarding the bioavailability of systemic oral curcumin used in RCTs. We searched PubMed on 12 September 2020, to find articles reporting RCTs that used curcumin as an intervention. We extracted data about trial characteristics, curcumin products used, methods for improving curcumin bioavailability, and mentions of curcumin bioavailability. We included 165 RCTs. The most common category of intervention was simply described as “curcumin” or “curcuminoids” without a commercial name. There were 107 (64%) manuscripts that reported that they used methods to enhance the oral bioavailability of curcuminoids used in their intervention; 25 different methods were reported. The most common method was the addition of piperine (23%). Phospholipidated curcumin, a combination of curcumin and turmeric oils, nanomicellar curcumin, and colloidal dispersion of curcumin were the next most common methods. Fourteen trials (8.4%) compared more than one different curcumin product; nine (7.9%) trials compared the bioavailability/pharmacokinetics of curcumin products. In conclusion, a high number of diverse methods were used, and very few trials compared different curcumin products. More studies are needed to explore the comparative bioavailability and efficacy of different curcumin products.

## 1. Introduction

Curcumin or diferuloylmethane [1,7-bis(4-hydroxy-3-methoxyphenyl)-1,6-heptadiene-3,5-dione] is a natural polyphenol isolated from the rhizome of turmeric (*Curcuma longa* L.), a member of the ginger family (Zingiberaceae). Numerous medicinal claims are associated with this popular spice [1]. However, drawbacks of curcumin include low bioavailability, poor tissue distribution, extensive metabolism, chemical instability, and potential for toxicity [2]. The main obstacle to its potential medical application is its insolubility in water, as the reported solubility of curcumin in deionized/distilled water at 25 °C is either 1.34 µg/mL [3] or 2.677 µg/mL [4].

Bioavailability is a term used to describe the percentage, or the fraction, of an administered dose of a xenobiotic that reaches the systemic circulation. Thus, bioavailability is essential in oral dosage form development [5]. Multiple studies have reported a relatively lower bioavailability after oral administration of curcumin [6,7]. Preclinical data and clinical studies on volunteers confirmed a small amount of absorption in intestines, hepatic first-pass effect, and a certain degree of intestinal metabolism as contributors to the poor systemic availability of curcumin when given orally [8].

It has been suggested that formulations that include adjuvants, nanoparticles, liposomes, micelles, and phospholipid complexes should improve bioavailability and enable longer circulation, better permeability, and resistance to curcumin’s metabolic processes [9].

A randomized controlled trial (RCT) is a prospective experimental study used to measure the efficacy of an intervention. In an RCT, participants are randomized into multiple arms, which receive different interventions and comparators. Participants are followed and compared. If they are designed adequately, it is considered that RCTs may achieve adequate control over confounding factors to enable an objective comparison of the interventions studied. However, as the RCTs of curcumin are accumulating, it is unknown how those trials approach the problem related to curcumin bioavailability and how it can impact trial results.

This study aimed to analyze whether RCTs of curcumin pay any attention to the bioavailability of systemic oral curcumin, whether trialists attempted to use methods that could improve this bioavailability, and whether they discussed their results in terms of curcumin bioavailability.

## 2. Results

The search yielded a total of 319 records, of which we excluded 154 for reasons reported in Appendix A. The remaining 165 studies were included in the analysis. The list and characteristics of included RCTs are reported in detail in Appendix A. Summary characteristics of included studies are shown in Table 1.

The included RCTs were published between the years 1980 and 2020. Trials were published by 112 journals, most commonly in the journal *Phytotherapy Research* (Table 1). Protocol registration was reported in 95 (58 %) studies; among those, most of the RCTs were registered on ClinicalTrials.gov (Table 1). The median number of participants randomized was 60. The median of two study arms were included. The median duration of patient follow-up was 4.5 weeks (Table 1).

The most common categories of participants included were healthy participants, followed by patients with arthritis, diabetes mellitus type 2, skin diseases, nonalcoholic fatty liver disease, and metabolic syndrome (Table 1). All categories of participants are shown in Appendix A.

Among included 165 trials, there were 184 curcumin interventions, as some trials used more than one curcumin product as a tested intervention. The most common category of curcumin intervention was simply described as “curcumin” or “curcuminoids” without a commercial name (Table 2).

A complete description of curcumin from methods reported in included RCTs (commercial name, details, dose, instructions for use) is reported in detail in Appendix A.

Among the 165 included RCTs, 112 (68%) mentioned the bioavailability of curcumin throughout the manuscript, regardless of the context. The parts of the manuscript where this was most commonly mentioned were the Introduction, Methods, and Discussion. Verbatim extractions of those parts of the text where the bioavailability of curcumin was mentioned are shown in Appendix A.

There were 106 (64%) manuscripts that reported the use of methods to enhance the oral bioavailability of curcuminoids used in their intervention. The trials reported 25 different methods for enhancing the bioavailability of oral curcumin. The most common method was the addition of piperine, used in 23% of the trials. Phospholipidated curcumin, a combination of curcumin and turmeric oils, nanomicellar curcumin, and colloidal dispersion of curcumin were the next most common methods (Table 3).

Most trials that used methods for enhanced oral bioavailability of curcumin were published in the most recent analyzed 5 years; only 12 trials using such methods were published before 2014.

All trials that mentioned bioavailability did not use methods to enhance the bioavailability, and vice versa. Among the 53 trials that did not mention bioavailability anywhere in the manuscript, 10 (19%) used methods to enhance curcumin bioavailability. Of 112 trials that mentioned bioavailability in the manuscript, 16 (14%) did not report the use of any methods to enhance curcumin bioavailability.

There were 14 (8%) trials that compared more than one type of standardized curcumin product. Most of them compared two curcumin products; while one compared three products, and two compared four products (Table 4). Nine of those fourteen trials compared the bioavailability or pharmacokinetics of the tested curcumin products. Trials that compared the enhanced versions with the standard curcumin reported that the bioavailability was higher with the enhanced products (Table 4). Six trials compared the effect of supplementation of various curcumin products in individuals with metabolic syndrome, osteoarthritis, or those suffering from experiencing occupational stress-related anxiety and fatigue. Results were heterogeneous, as half of those trials did not demonstrate any superiority of enhanced products on the investigated outcomes (Table 4).

## 3. Discussion

This study showed that 68% of RCTs that analyzed the effects of curcumin used methods for enhancing the oral bioavailability of oral curcumin. The most common enhancement method was the addition of piperine to curcumin. Most of the trials that used methods for enhancing the oral bioavailability of curcumin were published in the five most recent analyzed years. However, very few trials (7.9%) compared the bioavailability/pharmacokinetics of various curcumin products.

Oral bioavailability is a crucial aspect of the bio-efficiency of bioactive food ingredients, as it influences the potential health benefits of food [10]. However, many biocomponents have a low oral bioavailability because of their low stability in gastrointestinal fluid and inadequate absorption through the intestinal epithelium [11,12]. Thus, researchers have started studying the possibilities of improving the oral bioavailability of food bioactive ingredients by incorporating bioactive agents into different colloidal delivery systems such as emulsions, nanoemulsions, microemulsions, solid lipid nanoparticles, biopolymer nanoparticles, microgels, et cetera [6,13,14,15,16].

As curcumin has low aqueous solubility and can be metabolized rapidly by the gastrointestinal system, resulting in low oral bioavailability, multiple pharmaceutical strategies for oral administration of curcumin have also been tested, including solid dispersions, nano/microparticles, polymeric micelles, nanosuspensions, lipid-based nanocarriers, cyclodextrins, conjugates, and polymorphs [17].

It is reasonable to expect that different formulations of curcumin and different methods used to improve its oral bioavailability will impact human clinical trials. In a narrative review published in 2019, Ma et al. presented 39 trials that used curcumin in a tabular form, but without reporting methods for searching the literature and without a clear conclusion regarding the methods used or their comparative advantage [17].

In 2019, Kunnumakkara published a non-systematic analysis of “over 200 clinical studies with curcumin”, claiming that the “therapeutic potential of curcumin as demonstrated by clinical trials has overpowered the myth that poor bioavailability of curcumin poses a problem.” [18]. However, their study did not report any research methods, analysis methods, or eligibility criteria [18]. Such generic statements cannot be made based on non-systematic narrative literature reviews.

In this study, we analyzed all RCTs indexed until the targeted search date on PubMed that used curcumin to explore methods to enhance its oral bioavailability. In 107 trials that reported using such methods, 25 different methods were described. This is a widely heterogeneous methodological approach, and it is not clear what the comparative efficacy of all those different approaches is. Only 14 out of 165 trials we analyzed compared different curcumin products (from two to four products), and of those 14 trials, 9 compared their bioavailability/pharmacokinetics. Considering the vast heterogeneity of different methods described for enhancing the bioavailability of curcumin, the number of RCTs that have compared bioavailability/pharmacokinetics of different curcumin products is extremely low.

Moreover, six trials that compared different curcumin products have compared their effects on various clinical outcomes regarding efficacy and/or safety. Some of those studies did not find that products with methods that are supposed to enhance availability were better than the standard curcumin. This indicates that we not only need trials that will compare the bioavailability of different methods, but also trials that will compare different curcumin products head-to-head to test their comparative clinical efficacy.

Of note, we found that among 113 trials that mentioned bioavailability in the manuscript, 14% did not report using any methods to enhance curcumin bioavailability. This indicates that some trialists were aware of the issues related to curcumin bioavailability but still chose not to try to enhance it.

While the focus of this study was to analyze aspects related to the bioavailability of curcumin, we noticed poor reporting of investigated interventions. Some studies reported “generic” names or descriptions of the intervention, while some reported commercial names. Some studies reported both, but with an incomplete list or description of the investigated interventions. Furthermore, the trialists interchangeably used the terms curcumin and curcuminoids, even though they are not synonymous. Thus, we would urge trialists to devote more attention to detailed and correct reporting of the intervention that was used in a trial. Transparent reporting will foster the replicability of a trial.

Evidence syntheses from this field of research should compare the effects of different methods for enhancing the oral bioavailability of curcumin in data from clinical trials. Such analysis should involve data extraction on outcomes for efficacy and safety for different indications and performing comparative evaluations between trials that used different methods. However, that was beyond the scope of this study.

This study had several limitations. First, we searched only PubMed to retrieve RCTs on curcumin. Since this was not a systematic review, we used a single search system to retrieve comprehensive datasets of trials to analyze. A recent comparison of 26 different academic search systems that could be used in systematic reviews and meta-analyses showed that PubMed is suitable for use as a principal search system [19]. Second, we used very specific inclusion criteria. However, we provided detailed reasons for excluding the retrieved studies so that our decision-making is transparent.

## 4. Materials and Methods

### 4.1. Study Design

This was a methodological study in which we included RCTs that were indexed on PubMed.

### 4.2. Inclusion Criteria

We included RCTs where curcumin was used via oral ingestion for systemic absorption, and analyzed as an intervention or a comparator, regardless of the type of participants and type of outcomes that were used in a trial. We also included RCT protocols if they were retrieved via our search. We included manuscripts that reported post hoc analyses of previously published RCTs.

### 4.3. Exclusion Criteria

We excluded studies where mixtures of curcumin with other compounds were used as an intervention in a way that the intervention effects could not be attributed clearly to curcumin (for example, curcumin as a spice mixture in food, or as a part of herbal intervention). The only exception was interventions where curcumin was combined with a bioavailability enhancer. We excluded studies using turmeric powders. We excluded studies on animals.

We excluded studies where curcumin was used via administration modes other than oral ingestion for systemic absorption, such as curcumin taken in the mouth only as a local intervention in the oral cavity, or intravenous or rectal application of curcumin.

### 4.4. Search and Screening

We searched PubMed on 12 September 2020 by using the following search syntax: curcumin OR curcuma OR turmeric OR theracurmin OR tetrahydrocurcumin OR NCB-02 OR Curcuma domestica Val. OR Curcuma xanthorrhiza OR diferuloylmethane OR curcuminoids OR Biocurcumax OR biocurcumin OR BCM-95 OR BCM-095, and combined it with the limit for randomized controlled trials.

All bibliographic records that were found with this search were retrieved. Two authors independently screened all records to verify that they indeed fulfilled the inclusion criteria. We then retrieved full texts of all RCTs that were deemed eligible or potentially eligible and repeated screening of full texts against the inclusion criteria. Again, two authors independently screened each full text. Reasons for excluding records from the study were noted. In case of disagreement during the screening of bibliographic records and full texts, two authors resolved discrepancies in opinion via discussion, or if necessary, a third author was included in the decision-making.

### 4.5. Data Extraction

Two authors participated in data extraction for each study; one author conducted the extraction, and the second author verified the extraction. For RCTs retrieved via PubMed, we extracted the following information: the last name of the first author, year of publication, the title of study, trial registration number, number of participants randomized, number of study arms, duration of follow-up, type of participants included, type of curcumin product that was tested (either as an intervention or a comparator), and a complete description of curcumin from the methods. We also extracted any mentions about the bioavailability of curcumin throughout the manuscript, together with the manuscript section where considerations regarding curcumin bioavailability were mentioned. For the type of curcumin that was used, we extracted information for both commercial and generic names of the product, as described in the article.

After data extraction, we categorized the responses and presented the data narratively.

### 4.6. Statistics

Data were presented as numbers and frequencies, median, and range. Microsoft Excel (Microsoft Inc., Redmond, WA, USA) was used for analyses.

## 5. Conclusions

The majority of trials reported using methods for enhancing the oral bioavailability of curcumin. However, a large number of diverse methods were used, and very few trials compared different curcumin products. More studies are needed to explore the comparative bioavailability and efficacy of different curcumin products.

## Figures and Tables

**Table 1 pharmaceuticals-15-00939-t001:** Characteristics of included studies (*N* = 165).

Characteristics	Values
**The most common journals where trials were published, *N* (%)** *Phytotherapy Research* *Complementary Therapies in Medicine* *Journal of Clinical Psychopharmacology* *Journal of Dietary Supplements* *Drug Research (Stuttgart)* *Molecular Nutrition & Food Research*	21 (13) 6 (3.6) 4 (2.4) 4 (2.4) 3 (1.8) 3 (1.8)
**The most common registries where trials were registered, *N* (%)** Not reported ClinicalTrials.gov Iranian Registry of Clinical Trials (IRCT) Australian New Zealand Clinical Trials Registry UMIN Clinical Trials Registry	69 (42) 37 (22) 34 (21) 8 (4.8) 4 (2.4)
**Number of participants randomized, median (IQR)**	60 (33.5 to 88.5)
**Number of study arms, median (IQR)**	2 (2 to 2)
**Duration of patient follow-up in weeks, median (IQR)**	4.5 (0.375 to 10.5)
**The most common categories of participants included, *N* (%)** Healthy volunteers Patients with arthritis Patients with type 2 diabetes mellitus Patients with skin disease Patients with nonalcoholic fatty liver disease (NAFLD) Patients with metabolic syndrome	33 (20) 16 (9.6) 13 (7.8) 11 (6.6) 10 (6.0) 10 (6.0)

IQR = interquartile range.

**Table 2 pharmaceuticals-15-00939-t002:** The most common types of curcumin products used in analyzed trials (*N* = 165); interventions used in two trials or more are shown. Commercial names of the products are marked with an asterisk.

Curcumin Product Types	*N* (%)
Curcumin/curcuminoids	59 (36)
Components: Unformulated curcumin/curcumionoids	
Curcumin C3 Complex + Bioperine *	20 (12)
Components: Curcumin preparation containing the three major curcuminoids including curcumin, demethoxycurcumin, and bisdemethoxycurcumin in patented ratio + a patented extract from *Piper nigrum* standardized minimum to 95% piperine	
Meriva *	18 (11)
Components: Phospholipidated curcumin containing a complex of curcuminoids and soy phosphatidylcholine in a 1:2 weight ratio and two parts of microcrystalline cellulose	
BCM-95 *	15 (9.0)
Components: A proprietary combination of 95% curcuminoids and volatile oils from turmeric rhizome	
Theracurmin *	12 (7.2)
Components: Curcumin preparation produced by patented, colloidal dispersion technology	
Sinacurcumin *	11 (6.6)
Components: Curcuminoids nanomicelles	
Curcumin C3 Complex *	7 (4.2)
Components: Curcumin preparation containing the three major curcuminoids including curcumin, demethoxycurcumin, and bisdemethoxycurcumin in patented ratio	
Nanocurcumin	4 (2.4)
Components: Nanocurcumin	
Curcumin and piperine	3 (1.8)
Components: Unformulated curcumin and piperine	
CGM *	2 (1.2)
Components: Curcuminoides in form of curcumin–galactomannoside complex	
Longvida *	2 (1.2)
Components: Solid lipid particle formulation of curcumin	
NCB-02 *	2 (1.2)
Components: Standardized turmeric extract comprising 72% curcumin, 18.08% demethoxycurcumin, and 9.42% bisdemethoxycurcumin	
CurcuWin *	2 (1.2)
Components: Formulation of curcumin a hydrophilic carrier (63–75%), cellulosic derivatives (10–40%), and natural antioxidants (1–3%)	
Turmix *	2 (1.2)
Components: Formulation comprising *Curcuma longa* L. extract (98%) and *Piper nigrum* extract (2%)	

* Commercial names.

**Table 3 pharmaceuticals-15-00939-t003:** Methods used to enhance the oral bioavailability of curcuminoids in analyzed trials (*N* = 107).

Method	*N* (%)
Piperine	26 (24)
Phospholipidated curcumin	19 (18)
Turmeric oils	17 (14)
Nanomicellar curcumin	12 (11)
Colloidal dispersion of curcumin	12 (11)
Nanocurcumin	4 (3.7)
Curcumin(oids)–galactomannoside complex	2 (1.9)
Curcumin in a turmeric matrix formulation	2 (1.9)
Dispersion of curcumin and antioxidants on a water-soluble carrier	2 (1.9)
Solid-lipid particle formulation	2 (1.9)
Amorphous curcuminoid dispersion	1 (0.09)
Curcumin complexed with phosphatidylserine and piperine	1 (0.09)
Curcumin embedded in liposomal membranes	1 (0.09)
Curcumin embedded with surfactants, polar lipids, and solvents	1 (0.09)
Curcumin with a volatile oil	1 (0.09)
Curcuminoids blended with rhizome powder	1 (0.09)
Curcuminoids infused into fenugreek fiber	1 (0.09)
Curcuminoid micelles prepared with Tween-80	1 (0.09)
Curcumin–phospholipid complex	1 (0.09)
Gamma–cyclodextrin complex containing curcumin	1 (0.09)
Extraction from fresh turmeric	1 (0.09)
Microencapsulated curcumin	1 (0.09)
Micromicellar curcumin(oids) formulation	1 (0.09)
Nanoparticle-curcumin nanocrystals stabilized with a hydrophilic polymer	1 (0.09)

**Table 4 pharmaceuticals-15-00939-t004:** Studies that have compared multiple curcumin products. Commercial names are marked with an asterisk.

Study PMID	Products Compared	Study Aim	Conclusions Regarding Bioavailability (Verbatim Quotes)
30020812	Curcumin Meriva *	To investigate the effects of curcumin on serum copper (Cu), zinc (Zn), and Zn/Cu ratio levels in patients with metabolic syndrome.	Serum Zn concentration was increased significantly in the phospholipidated curcumin and curcumin groups after intervention, and it was significantly higher (*p* < 0.001) in the phospholipidated curcumin group than in the curcumin group (*p* < 0.05). The effect of phospholipidated curcumin on zinc was higher than the effect of curcumin because phospholipidated curcumin has better bioavailability than curcumin.
29958053	Curcumin Meriva *	To investigate the effects of unformulated curcumin and phospholipidated curcumin on antibody titers to heat shock protein 27 (anti-Hsp 27) in patients with metabolic syndrome (MetS).	Study used phospholipidated curcumin, which is known to be more bioavailable compared to unformulated curcumin, but no significant changes in serum anti-Hsp 27 and anthropometric measures in patients with MetS following supplementation could be found.
29974228	Lipisperse * Curcumin	To investigate the pharmacokinetics of a commercially available curcumin extract, with or without the curcumin–LipiSperse^®^ delivery complex.	The novel delivery system LipiSperse ^®^ is safe in humans, and demonstrates superior bioavailability for the supply of curcumin when compared to a standard curcumin extract.
29043927	BioCurc * Curcumin	To assess the bioavailability of a novel curcumin formulation compared to 95% curcumin and published results for various other curcumin formulations.	The novel curcumin liquid droplet micromicellar formulation (CLDM) formulation facilitates absorption and produces exceedingly high plasma levels of both conjugated and total curcumin compared to 95% curcumin.
29316908	CuraMed (BCM-95) * Curamin *	To assess the efficacy and safety of curcuminoid complex extract from turmeric rhizome with turmeric volatile oil (CuraMed^®^) and its combination with boswellic acid extract from Indian frankincense root (Curamin^®^) vs. placebo for the treatment of 40- to 70-year-old patients with osteoarthritis (OA).	Twelve-week use of curcumin complex or its combination with boswellic acid reduces pain-related symptoms in patients with OA. Curcumin in combination with boswellic acid is more effective. Combining *Curcuma longa* and *Boswellia serrata* extracts in Curamin^®^ increases the efficacy of OA treatment, presumably due to synergistic effects of curcumin and boswellic acid.
28840615	Curcumin Meriva *	To investigate the effect of curcumin supplementation on the serum pro-oxidant-antioxidant balance (PAB) in patients with MetS.	Serum PAB increased significantly in the curcumin group (*p* < 0.001), but in the phospholipidated curcumin group, elevation of PAB level was not significant (*p* = 0.053). The results of our study did not suggest any improvement of PAB following supplementation with curcumin in MetS subjects.
28198120	Curcumin-phospholipid complex Curcumin	To investigate the effect of curcumin on serum vitamin E levels in subjects with MetS.	Results of the present study did not suggest any improving effect of curcumin supplementation on serum vitamin E concentrations in subjects with MetS.
27043120	CGM Curcumin	To investigate the safety, antioxidant efficacy, and bioavailability of CurQfen (curcumagalactomannoside [CGM]), a food-grade formulation of natural curcumin with fenugreek dietary fiber that has been shown to possess improved blood–brain barrier permeability and tissue distribution in rats.	The study demonstrated the safety, tolerance, and enhanced efficacy of CGM in comparison with unformulated standard curcumin. Further comparison of the free curcuminoids bioavailability after a single-dose (500 mg once per day) and repeated-dose (500 mg twice daily for 30 days) oral administration revealed enhanced absorption and improved pharmacokinetics of CGM upon both single- (30.7-fold) and repeated-dose (39.1-fold) administrations.
24461029	Meriva * BCM95 * CurcuWin * C3 complex *	To comparatively measure increases in levels of curcuminoids (curcumin, demethoxycurcumin, bisdemethoxycurcumin) and the metabolite tetrahydrocurcumin after oral administration of three different curcumin formulations in comparison to unformulated standard.	A formulation of curcumin with a combination of hydrophilic carriers, cellulosic derivatives, and natural antioxidants significantly increases curcuminoid appearance in the blood in comparison to unformulated standard curcumin CS, CTR, and CP.
22401804	Microencapsulated curcumin Curcumin	To investigate the human bioavailability of curcumin from breads enriched with 1 g/portion of free curcumin (FCB), encapsulated curcumin (ECB), or encapsulated curcumin plus other polyphenols (ECBB) was evaluated.	Curcuminoid encapsulation increased their bioavailability from enriched bread, probably preventing their biotransformation, with combined compounds slightly reducing this effect.
21413691	Meriva * Curcumin	To investigate the relative absorption of a standardized curcuminoid mixture and its corresponding lecithin formulation (Meriva) in a randomized, double-blind, crossover human study.	The improved absorption, and possibly also a better plasma curcuminoid profile, might underlie the clinical efficacy of Meriva at doses significantly lower than unformulated curcuminoid mixtures.
28204880	Curcumin Gamma–cyclodextrin complex containing curcumin Meriva * BCM95 *	To investigate the bioavailability of a new γ-cyclodextrin curcumin formulation (CW8). This formulation was compared to a standardized unformulated curcumin extract (StdC) and two commercially available formulations with purported increased bioavailability: a curcumin phytosome formulation (CSL), and a formulation of curcumin with essential oils of turmeric extracted from the rhizome (CEO).	The data presented suggest that γ-cyclodextrin curcumin formulation (CW8) significantly improves the absorption of curcuminoids in healthy humans.
27503249	Meriva * C3 complex *	To evaluate the relationship between steady-state plasma and rectal tissue curcuminoid concentrations using standard and phosphatidylcholine curcumin extracts in a randomized, crossover study	When adjusting for curcumin dose, tissue curcumin concentrations were five-fold greater for the phosphatidylcholine extract. Improvements in curcuminoid absorption due to phosphatidylcholine are not uniform across the curcuminoids. Furthermore, curcuminoid exposures in the intestinal mucosa are most likely due to luminal exposure rather than plasma disposition. Finally, once-daily dosing is sufficient to maintain detectable curcuminoids at a steady state in both plasma and rectal tissues.
29027274	Cureit/Acumin * Curcu-Gel * Doctor’s Best Curcumin Phytosome *	To assess the bioavailability of a completely natural turmeric matrix formulation (CNTMF) and compare its bioavailability with two other commercially available formulations, namely, curcumin with volatile oil (volatile oil formulation) and curcumin with phospholipids and cellulose (phospholipid formulation), in healthy human adult male subjects (15 each group) under fasting conditions.	The results of this study indicate that curcumin in a natural turmeric matrix exhibited greater bioavailability than the two comparator products.

Acronyms: CEO = curcumin with essential oils of turmeric extracted from the rhizome; CGM = curcumagalactomannoside; CLDM = curcumin liquid droplet micromicellar formulation; CNTMF = completely natural turmeric matrix formulation; CP = curcumin phytosome formulation; CS = standardized curcumin mixture; CSL = curcumin phytosome formulation; CTR = curcumin formulation with volatile oils of turmeric rhizome; CW8 = γ-cyclodextrin curcumin formulation; MetS = metabolic syndrome; OA = osteoarthritis; PAB = pro-oxidant-antioxidant balance; StdC = standardized unformulated curcumin extract. * Commercial names.

## Data Availability

Data are contained within the article and Appendix A.

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
