# Peer review of "Methods Used for Enhancing the Bioavailability of Oral Curcumin in Randomized Controlled Trials: A Meta-Research Study"

_pharmaceuticals, 2022, doi:10.3390/ph15080939_

Round 1

Reviewer 1 Report

This manuscript by Mimica et al. investigated on the enhancing the bioavailability of oral curcumin is straightforward and attractive. The obstacle of the curcumin application is the bioavailability. The authors analyzed methods and reporting regarding bioavailability of systemic oral curcumin used in randomized controlled trials. The results can attract a lot of audience.

Comments

A.   In the work, many brands curcumin were applied. Please introduce the components or the formula briefly.

Randomized controlled trials are the main method of the work. Please describe some words about the trials.

Author Response

Response to Reviewer 1

Dear Reviewer 1,

We are grateful for Your time and effort invested in providing feedback for our manuscript.

Based on the feedback, we made the following revisions:

  1. Moderate English changes required

Author response: We conducted language editing and made further revisions.

  1. In the work, many brands curcumin were applied. Please introduce the components or the formula briefly.

Author response: We have added these details in the main manuscript in Table 2. Previously, we indicated the following in the manuscript:

“A complete description of curcumin from methods reported in included RCTs (commercial name, details, dose, instructions for use) is reported in detail in Supplementary table 2.”

The Supplementary table 2 still has all this information.

  1. Randomized controlled trials are the main method of the work. Please describe some words about the trials.

Author response: We have now expanded the text to describe what are randomized controlled trials. If the Reviewer wanted to suggest that we provide more details about the included trials, these are provided in detail in the Supplementary file 2.

We hope that the revised manuscript will be acceptable.

Sincerely,

The authors

Reviewer 2 Report

Dear Authors,

The paper presents a literature synthesis regarding curcumin bioavailability studies in RCT.  It is interesting the selection of studies that compared multiple curcumin products. 

The article is well realized.

Author Response

Response to Reviewer 2

Dear Reviewer 2,

We are grateful for Your time and effort invested in providing feedback for our manuscript.

The Reviewer commented:

The paper presents a literature synthesis regarding curcumin bioavailability studies in RCT.  It is interesting the selection of studies that compared multiple curcumin products. 

The article is well realized.

Author response: We are grateful for the kind words of the Reviewer 2 regarding our manuscript.

Sincerely,

The authors

Reviewer 3 Report

The conclusion section must be improved, it reads inconclusive! Authors can comment on things such as which method was the most successful with curcumin bioavailability enhancement, etc.

Author Response

Response to Reviewer 3

Dear Reviewer 3,

We are grateful for Your time and effort invested in providing feedback for our manuscript.

Based on the feedback, we made the following revisions:

  1. Moderate English changes required

Author response: We conducted language editing and made further revisions.

  1. The conclusion section must be improved, it reads inconclusive! Authors can comment on things such as which method was the most successful with curcumin bioavailability enhancement, etc.

Author response: We appreciate the Reviewer’s suggestion. However, comparisons between efficacy of the methods were not the goal of this study. We have explicitly stated this in the manuscript, just above the conclusions:

“Evidence syntheses from this field of research should compare the effects of different methods for enhancing the oral bioavailability of curcumin in data from clinical trials. Such analysis should involve data extraction on outcomes for efficacy and safety for different indications and performing comparative evaluations between trials that used different methods. However, that was beyond the scope of this study.”

We made further revisions of the manuscript, based on the suggestions of other Reviewers.

We hope that the revised manuscript will be acceptable.

Sincerely,

The authors